# DeepScaleR: Effective RL Scaling of Reasoning Models via Iterative Context Lengthening

## Abstract

Recent advances in large reasoning models (LRMs) such as OpenAI's o1 and Deepseek-R1 have demonstrated that reinforcement learning (RL) with outcome-based supervision can significantly enhance the reasoning abilities of language models. However, these improvements have so far relied on massive model scales and compute budgets, leaving open the question of whether RL-based scaling can be made both effective and efficient at smaller scales. In this work, we introduce DeepScaleR-1.5B, a 1.5B parameter model trained using reinforcement learning via a novel iterative context lengthening strategy. Our method begins with shorter context windows and progressively extends them throughout training, enabling the model to first learn to reason efficiently before learning to reason longer. This approach yields substantial performance gains with dramatically reduced computational cost. DeepScaleR-1.5B achieves 43.3% Pass@1 on the AIME2024 math benchmark—a 14.3 percentage point improvement over its base model and on par with OpenAI's o1-preview—while requiring a fraction of the compute. We provide a full training recipe, including dataset, code, hyperparameters, and training methodology, demonstrating that small models can be effectively scaled into strong math reasoners via RL.

## 1 Introduction

The release of OpenAI o1 [29] and Deepseek-R1 [12] marks a paradigm shift in improving the reasoning capabilities of large language models. These models, also known as large reasoning models (LRMs), achieve remarkable performance on challenging reasoning tasks such as competition-level mathematics and coding—far surpassing the capabilities of traditional, non-reasoning models. Unlike standard models, LRMs are explicitly trained to "think longer" by leveraging extended context during inference to arrive at correct and well-reasoned conclusions. This enables them to outperform conventional LLMs by a substantial margin.

Many approaches have been discussed and explored to encourage models to make more extensive use of the context before committing to a final answer. Some early training-free approaches leverage prompting techniques to ask the model to think step by step [20]. Later, many works perform supervised fine-tuning on long CoT trajectories curated through either distillation [28, 23] or expert written trajectories [52].

Beyond prompting and supervised finetuning, the recent release of Deepseek-R1 [12] demonstrates that reinforcement learning (RL) with outcome-based rewards can be surprisingly effective in enhancing a model's reasoning ability. Notably, Deepseek-R1 shows that by directly supervising solution correctness, the model naturally learns to "think longer"—leveraging extended context before producing an answer. As training progresses, the model's average response length increases organically, reflecting a growing tendency toward more deliberative reasoning.

While Deepseek-R1 lays out a high-level blueprint demonstrating the potential of RL training with outcome supervision, it leaves critical details undisclosed, including the dataset, hyperparameters, and scaling methodology. Moreover, training such a large model is prohibitively expensive—Deepseek-R1 is a 671B MoE model trained over 8,000 steps. This raises an important open question: can RL-based reasoning improvements be scaled effectively to smaller models under realistic compute constraints? Initial results from Deepseek-R1 [12] suggest that scaling down is not effective. When applied to the

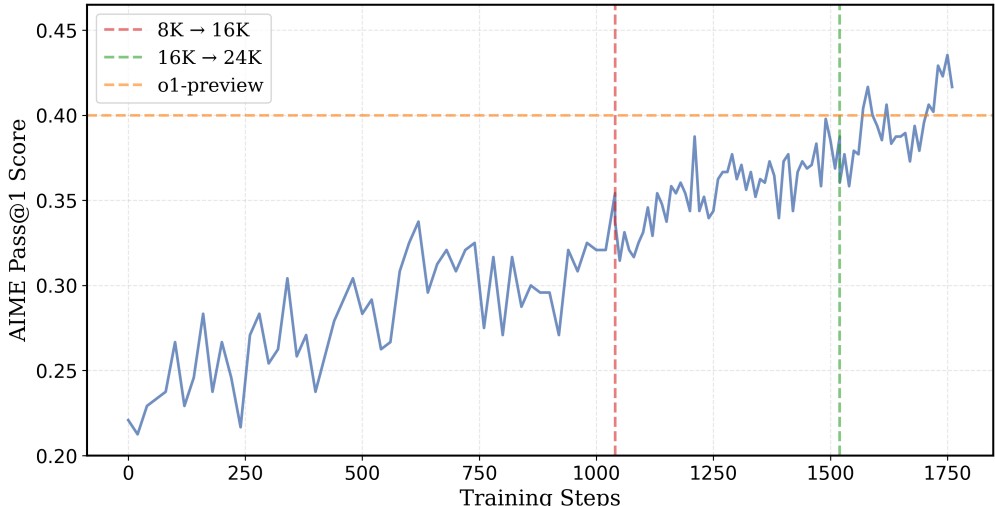

Figure 1: DeepScaleR's Pass@1 accuracy on AIME2024 as training progresses. At step 1040 and 1520, the context length is extended to 16K and 24K.

32B Qwen model, performance on the AIME competition math dataset reached only 47%, compared to the 79% achieved by R1, indicating diminishing returns at smaller scales.

In addition, even with smaller models, RL training remains computationally expensive. The primary challenges in scaling RL for reasoning models are:

1. **Long context lengths**: Reasoning tasks often require extended contexts—up to 32K tokens— unlike traditional workloads where outputs are typically only a few hundred tokens. This dramatically slows response generation and introduces major performance bottlenecks.

2. **Large batch sizes and prolonged training**: To achieve significant performance improvements (e.g., >10% Pass@1 on AIME), RL training demands thousands of gradient updates. Stability in training typically requires large batch sizes (e.g., 1024 rollouts per batch), making each training step extremely costly.

Given these factors, naively applying RL to train reasoning models at scale is impractical. For instance, we estimate that training even a modest 1.5B parameter model over 2,000 steps with a 32K context window requires approximately 17,500 A100 GPU hours or 21K USD in compute cost.

Thus, two key open questions remain: (1) *How can we effectively scale RL training to improve reasoning ability?* and (2) *How can we efficiently scale RL training to make it accessible under practical computational budgets?*

In this work, we answer both questions affirmatively. First, we show that RL scaling can be highly effective even for a small 1.5B model. Our model, **DeepScaleR-1.5B-Preview**, achieves 43.3% on AIME2024—an absolute improvement of 14.3% over the base model—and matches the performance of OpenAI's o1-preview through RL scaling alone. While Deepseek-R1's results suggested that direct RL scaling might be ineffective for smaller models, our work demonstrates that, with high-quality data distillation over long reasoning trajectories, small distilled models can be effectively transformed into strong reasoner using RL.

To address computational challenges and make RL training efficient, we propose *iterative context lengthening*, a simple yet highly effective strategy that first encourages the model to think shorter and more efficiently, before progressively "thinking longer" as the training evolves. Intuitively, our techniques acts as a implicit curriculum that forces the model to solve easier problems first with shorter, more efficient reasoning. Then, as training plaeteaus, we increase the context length to give the model more thinking space to solve harder problems.

Concretely, we adopt a three-stage training process: starting with an 8K context window, and later expanding to 16K and 24K contexts. During the initial 8K phase, the model's average response

length shrinks dramatically—from 16.3K tokens to 5.8K tokens on the AIME2024 dataset—while still gaining 5% in accuracy. This indicates that the model learns to reason better and more efficiently early on. As we gradually increase the context length to 16K and then 24K, the model continues to improve by "thinking longer", reaching 38% and 43% Pass@1 respectively, and ultimately matching o1-preview's competition math performance despite being much smaller in scale.

Furthermore, our training method dramatically improves computational efficiency. Our full training run requires only **3,800 A100 GPU hours** over 1,750 steps—a **2.6x×** reduction compared to the naive baseline of training 1,750 steps at 24K context directly. At inference time, our model achieves 50% majority-vote accuracy on AIME2024 using 14K fewer tokens per problem than the base model, demonstrating significantly more efficient test-time scaling.

In this work, we contribute the following:

- We propose *iterative context lengthening*, a simple yet effective technique that progressively extends the context length during RL training, enabling both more efficient training and stronger test-time scaling performance.

- Using *iterative context lengthening*, we train DeepScaleR-1.5B, a model that achieves significant performance gains in math reasoning through RL scaling, surpassing o1-preview results with a model orders of magnitude smaller, and provide the full training recipe.

- We study the effect of different context length schedules, propose general principles for selecting an optimal schedule, and empirically validate them through ablation experiments.

## 2 RELATED WORK.

**LLM reasoning**   A substantial body of work has explored bootstrapping and enhancing the mathematical and general reasoning capabilities of language models through prompting [48, 19, 60, 6, 51], inference-time scaling [39, 4, 36], and training-based approaches [8, 11, 16, 25, 31, 59, 49, 28, 52, 23]. Wei et al.[48] introduced chain-of-thought (CoT) prompting, which encourages models to "think step by step," revealing latent reasoning capabilities. Following the release of o1[29], a wave of work [39, 4] has focused on inference-time scaling, where multiple solutions are sampled and aggregated via majority voting or LLM-based verification. Beyond prompting and inference-time strategies, numerous studies investigate training methods to directly instill reasoning skills into models. For example, early works [43, 25] propose training a process reward model to guide solution search in mathematical problem-solving. Zelikman et al.[54] introduce rejection fine-tuning with self-generated rationales to bootstrap reasoning capabilities, inspiring several follow-up works that refine and extend this training paradigm [16, 55]. Other approaches integrate Monte Carlo Tree Search (MCTS) with process reward models for both training and inference [11, 59, 58, 49, 31], demonstrating that joint optimization across search, verification, and learning can enhance model reasoning performance.

**Reinforcement learning for LLMs**   The most widely adopted application of reinforcement learning in language models is Reinforcement Learning from Human Feedback (RLHF) [7, 30, 3, 62], which involves training a reward model from human preference data and using it to guide the model toward generating responses that are more aligned with human preferences. While RLHF originally uses PPO [34], some recent work proposes alternative methods (e.g. RLOO [1], Remax [24], Reinforce++ [18]) that removes the value model for more efficient RLHF training.

Beyond RLHF, a growing body of work [57, 2, 32, 5, 61] explores applying reinforcement learning to train LLMs for a range of decision-making tasks, including Android device control [2], web navigation and interaction [32], and text-based games [5, 61]. In contrast to RLHF, which is typically applied in a single-turn setting, these works operate in multi-turn environments, where standard policy gradient methods such as PPO [34] and REINFORCE [40] often suffer from sample inefficiency. As a result, many of these efforts explore off-policy or offline reinforcement learning methods [38, 32, 2] to improve training stability and data efficiency.

A parallel line of research applies reinforcement learning to enhance mathematical reasoning in LLMs [13, 37, 45, 35, 9, 21]. These methods typically leverage math datasets with verifiable rewards and either introduce new RL algorithms—such as GRPO [37] and PRIME [9]—or propose new

formulations for applying reinforcement learning in this domain [21]. Our work is along this line of research, showing that iterative context lengthening can be effective at scaling RL for math reasoning.

## 3 TRAINING RECIPE

In this section, we describe the methodology used to train DeepScaleR. Section 3.1 details the training setup, including the dataset, the reward function, and the reinforcement learning algorithm. Section 3.2 introduces our iterative context lengthening technique and presents the training procedure that enabled the model to reach o1-preview level performance on math reasoning tasks.

### 3.1 TRAINING SETUP

**Dataset curation**    We curate our training data from high-quality competition math problems, including AIME (1984–2023) [44], AMC (pre-2023), OMNI-MATH [10], and STILL3 [42]. To ensure reliable supervision, we implement a three-stage preprocessing pipeline: (1) **Answer extraction** — using `gemini-1.5-pro-002` [41] to parse official AoPS solutions; (2) **Duplicate removal** — applying retrieval-augmented generation with `all-MiniLM-L6-v2` [46, 33] to eliminate near-duplicates (>0.9 similarity) and prevent train–test contamination; (3) **Filtering** — excluding problems ungradable by `sympy` [27] to avoid noisy rewards. The final dataset contains **40K** unique problem–answer pairs.

**Reward function**    Following DeepSeek-R1, we use outcome-based rewards from ground-truth solutions: `1` if the answer is correct and well-formatted (LaTeX + `sympy` checks), otherwise `0`.

**Training algorithm**    We adopt Group Relative Policy Optimization (GRPO) [37, 12]. For question–answer pairs $(q, a)$, GRPO samples $G$ responses $\{o_i\}$ with rewards $\{r_i\}$ and optimizes:

$$\mathcal{J}_{\text{GRPO}}(\theta) = \mathbb{E}\left[\frac{1}{G}\sum_{i=1}^{G}\frac{1}{|o_i|}\sum_{t=1}^{|o_i|}\Big(\min\big(r_{i,t}(\theta)\hat{A}_{i,t}, \text{clip}(r_{i,t}(\theta), 1-\epsilon, 1+\epsilon)\hat{A}_{i,t}\big) - \beta D_{\text{KL}}(\pi_\theta||\pi_{\text{ref}})\Big)\right],$$

where

$$r_{i,t}(\theta) = \frac{\pi_\theta(o_{i,t}|q_i, o_{i,<t})}{\pi_{\theta_{\text{old}}}(o_{i,t}|q_i, o_{i,<t})}, \quad \hat{A}_{i,t} = \frac{r_i - \text{mean}(\{r_i\})}{\text{std}(\{r_i\})}.$$

**Base model**    We initialize from `DeepSeek-R1-Distilled-Qwen-1.5B` [12], distilled from `DeepSeek-R1` onto `Qwen2.5-Math-1.5B` [50], which improves math reasoning through extended reasoning tokens.

**Training hyperparameters**    Hyperparameters are provided in Appendix A.1.

### 3.2 ITERATIVE CONTEXT LENGTHENING

A key challenge in scaling RL for reasoning tasks is selecting an appropriate context window. Unlike standard RLHF, reasoning tasks often require very long outputs—for example, AIME solutions can exceed 10,000 tokens—creating a bottleneck for on-policy algorithms like GRPO, which must generate full trajectories before gradient updates. Autoregressive LLM generation with long contexts slows trajectory sampling and overall training.

This creates a fundamental trade-off: longer contexts allow tackling harder problems but increase computation, while shorter contexts improve efficiency but may limit reasoning. Iterative context lengthening addresses this by initially encouraging the model to "think shorter" with a constrained context window, then gradually increasing it to unlock longer-horizon reasoning. Our approach begins with RL training using an 8K context for efficient, effective reasoning, then incrementally expands to 16K and 24K to handle more challenging problems.

We next detail the training dynamics in each stage.

**Bootstrapping reasoning with an 8K context**   Before full-scale training, we did a diagnostic evaluation of `Deepseek-R1-Distilled-Qwen-1.5B` on AIME2024. The results from Table 1 incorrect responses were over three times longer than correct ones (20,346 vs. 6,395 tokens). This suggests that direct scaling at longer context might be inefficient, as these wrong responses are harder for the model to solve.

Therefore, we initialize training with an 8K context, providing an implicit curriculum that encourages concise reasoning on simpler problems and accelerates learning. While initial accuracy drops from 28.9% to 22.9%, training rewards steadily rise from 46% to 58%, and mean response length falls from 5,500 to 3,500 tokens. After 1K steps, DeepScaleR gains 5 points over the base model and 11 points compared to direct 8K training, while average response length shrinks from 16.3K to 5.8K tokens. This bootstrapping phase improves both performance and efficiency, making subsequent extended-context training substantially more tractable.

| Metric | Base Model | DeepScaleR-1.5B-8K | Change |
|---|---|---|---|
| AIME Pass@1 (%) | 28.9 | 33.9 | 5.0 |
| Avg. tokens (correct) | 6396.0 | 3661.2 | $-2734.8$ |
| Avg. tokens (incorrect) | 20 346.3 | 6976.8 | $-13 369.5$ |
| Avg. tokens (overall) | 16 335.6 | 5850.9 | $-10 484.7$ |

Table 1: Comparison of base model and 8K-context fine-tuned model on AIME2024. Training under constrained output length improves both efficiency and accuracy.

**Transitioning to 16K contexts**   After 1,000 training steps at 8K, response lengths began increasing, indicating the model was attempting longer reasoning. However, accuracy plateaued, rewards fluctuated, and the response clipping ratio rose from 4.2% to 6.5%, signaling that the 8K window was limiting further gains (See Figure 2 and 3).

Identifying this as a natural transition point, we checkpointed at step 1,040 and resumed training with a 16K context. This two-stage approach is more efficient than starting at 16K, as the 8K bootstrapping kept average response length at 3,500 instead of 10,000 tokens, reducing computation 2–3×. Following the switch, rewards, response length, and AIME accuracy steadily improved: after 500 steps, average response length rose to 5,500 tokens and Pass@1 accuracy reached 38

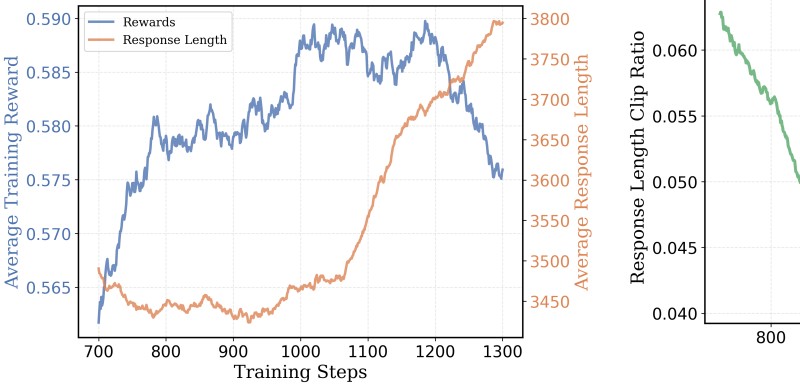
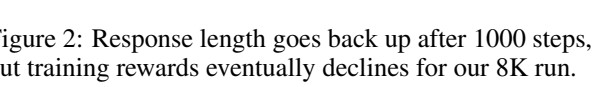
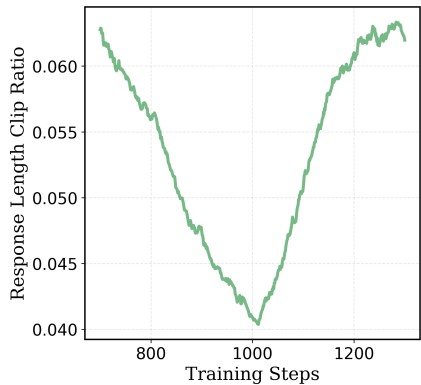

Figure 2: Response length goes back up after 1000 steps, but training rewards eventually declines for our 8K run.

Figure 3: The response length clip ratio rises after 1000 steps for the 8K context run.

**Final push with 24K contexts**   After an additional 500 steps at 16K context, performance once again began to plateau. Training rewards stabilized at 62.5%, AIME accuracy hovered at 38%, and output lengths began declining slightly. The maximum clipping ratio also rose to 2.0%, indicating renewed constraints at the new context ceiling.

To address this, we extended the context window one final time to 24K tokens, resuming training from step 480 of the 16K run. The results were immediate: within 50 steps, the model surpassed 40% accuracy, eventually reaching 43% by step 200 and surpassing `o1-preview`.

**Training efficiency and cost**   Overall, our training run consists of 1,750 steps. The initial 8K phase was trained on 8 A100 GPUs, while the 16K and 24K phases scaled up training to 32 A100 GPUs. In total, the training took around 3,800 A100 hours, equivalent to roughly 5 days on 32 A100s and $4500 in terms of compute cost.

### 3.3 PRINCIPLES FOR SELECTING THE CONTEXT WINDOW IN ITERATIVE LENGTHENING

While iterative context lengthening is an effective strategy for scaling reasoning models, it introduces a new hyperparameter: the context window. This raises two natural questions for model training: (1) what is the optimal initial context window, and (2) when should the window be expanded during training? Given our observations from DeepScaleR, we propose the principles for selecting context windows in iterative lengthening:

**Principle 1: Start at steepest gains**   Model performance vs. context length often follows a concave curve: rapid early gains plateau as context grows. We recommend starting fine-tuning near where initial gains taper, letting the model leverage short- to medium-length contexts before expanding. For example, on AIME2024 with `DeepSeek-R1-Distill-1.5B`, fixed-context Pass@1 scores at 2K, 4K, 8K, 16K, and 32K tokens are 3%, 9%, 23%, 26%, and 29%, showing steep gains up to 8K and diminishing returns afterward. This motivates a staged schedule of 8K → 16K → 24K. Conversely, if gains rise sharply only near the maximum context, direct training at the target length may be more effective than staged growth.

**Principle 2: Expand when performance plateaus**   Performance saturation, often accompanied by longer responses and higher clipping, indicates the model is constrained by context. Expanding the window before this plateau allows the model to fully utilize its reasoning capacity.

**General methodology**   Based on the above principles, iterative context lengthening can be implemented as a three-stage process: 1) evaluate coverage (fraction of problems solved) across context cutoffs; 2) identify the concave trend and select an initial window just beyond the steepest gains; 3) when coverage plateaus while response lengths increase, expand the context using the best checkpoint before the plateau. In Section 4.2, we present additional RL scaling experiments on the countdown task that empirically validate these principles.

## 4 EVALUATION

**Evaluation setup**   We evaluate our model on various competition-level mathematics benchmarks, including **AIME 2024** [44], **AMC 2023**, **MATH-500** [15], **Minerva Math** [22], and **Olympiad-Bench** [14]. Since datasets such as AIME has high variance, for each question, we sample 16 times following the recommended setup by Deepseek-R1 (temperature=0.6, topp=0.95) and report the average Pass@1 accuracy over the 16 trials.

We compare DeepScaleR with the base DeepSeek model and recent academic works exploring RL for math reasoning, including rStar [11], SimpleRL [56], PRIME [9], and STILL-3 [42]. We show our evaluation results in Table 2 and underline the model whose scores we evaluate and verify ourselves.

As shown in Table 2, DeepScaleR significantly outperforms the base model across all benchmarks, achieving a **14.4%** absolute gain on AIME2024 and an **8.1%** overall improvement. Additionally, DeepScaleR surpasses recent works such as rSTAR, PRIME, and SimpleRL, which are finetuned from a larger 7B models.

### 4.1 ABLATION STUDY

**Iterative Context Lengthening (8K → 16K → 24K) vs. Direct RL Scaling (24K)**   To evaluate the effectiveness of our iterative context lengthening strategy, we conduct an ablation study comparing

| Model | AIME 2024 | MATH 500 | AMC 2023 | Minerva Math | OlympiadBench | Avg. |
|---|---|---|---|---|---|---|
| Qwen-2.5-Math-7B-Instruct | 13.3 | 79.8 | 50.6 | 34.6 | 40.7 | **43.8** |
| rStar-Math-7B | 26.7 | 78.4 | 47.5 | - | 47.1 | **-** |
| Eurus-2-7B-PRIME | 26.7 | 79.2 | 57.8 | 38.6 | 42.1 | **48.9** |
| Qwen2.5-7B-SimpleRL | 26.7 | 82.4 | 62.5 | **39.7** | 43.3 | **50.9** |
| DeepSeek-R1-Distill-Qwen-1.5B | 28.8 | 82.8 | 62.9 | 26.5 | 43.3 | **48.9** |
| Still-3-1.5B-Preview | 32.5 | 84.4 | 66.7 | 29.0 | 45.4 | **51.6** |
| **DeepScaleR-1.5B-Preview** | **43.1** | **87.8** | **73.6** | 30.2 | **50.0** | **57.0** |
| O1-Preview | 40.0 | 81.4 | - | - | - | **-** |

Table 2: Pass@1 accuracy across competition-level math benchmarks. DeepScaleR outperforms both the base model and recent RL-enhanced methods.

our staged training approach (8K → 16K → 24K) against direct reinforcement learning (RL) scaling with a 24K context window. For the direct scaling baseline, we replicate the training configuration used in DeepScaleR's final 24K-stage run.

The direct 24K model is trained for 440 steps on 16 A100 GPUs, with each step taking approximately 1,300 seconds, amounting to a total training cost of roughly 2,400 A100 GPU-hours. To provide a fair comparison, we plot both training curves using GPU hours as the x-axis.

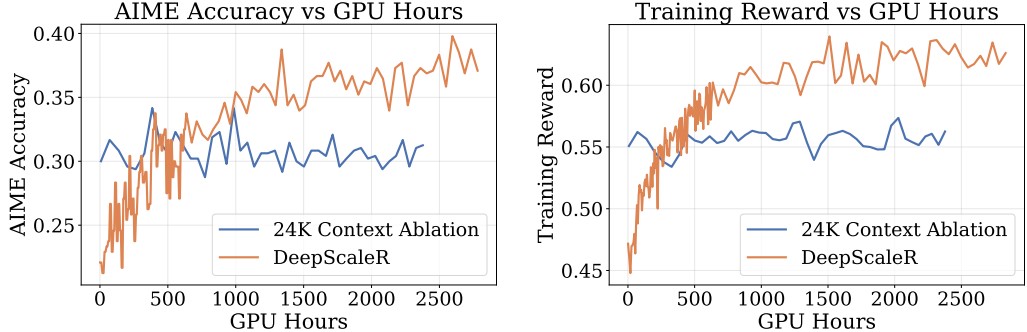

Figure 4: Comparison of DeepScaleR's iterative context lengthening (8K → 16K → 24K) versus direct RL scaling at 24K. **Left:** AIME accuracy vs. GPU hours. **Right:** Average training rewards vs. GPU hours.

As shown in Figure 4, the direct 24K scaling baseline exhibits unstable performance and does not yield significant improvement over time. In contrast, iterative context lengthening—while starting from a lower initial AIME accuracy due to truncation—demonstrates steady progress throughout training and surpasses the direct scaling baseline after approximately 500 GPU hours. These findings support our hypothesis that naively scaling to long contexts in RL training is suboptimal, and that a staged curriculum enables more stable and effective learning.

**Test-Time Scaling of DeepScaleR**    Test-time scaling refers to techniques that improve model performance on downstream tasks by allocating additional compute during inference. A widely adopted method is self-consistency [47], which generates multiple solutions and selects the final answer via majority voting.

Figure 5 presents a side-by-side comparison of test-time scaling between DeepScaleR and the original base model, `Deepseek-R1-Distill-1.5B`. For each of the 30 problems in the AIME2024 dataset, we generate 64 solutions per model and evaluate majority voting accuracy by repeatedly sampling subsets of responses from this pool. We run 300 sampling trials and report the mean accuracy and standard deviation as a function of the number of sampled solutions (left) and total number of generated tokens (right).

The results show that DeepScaleR consistently outperforms the base model as the number of samples increases, achieving a Maj@64 accuracy of 65% compared to 57.7%. Notably, our iterative context

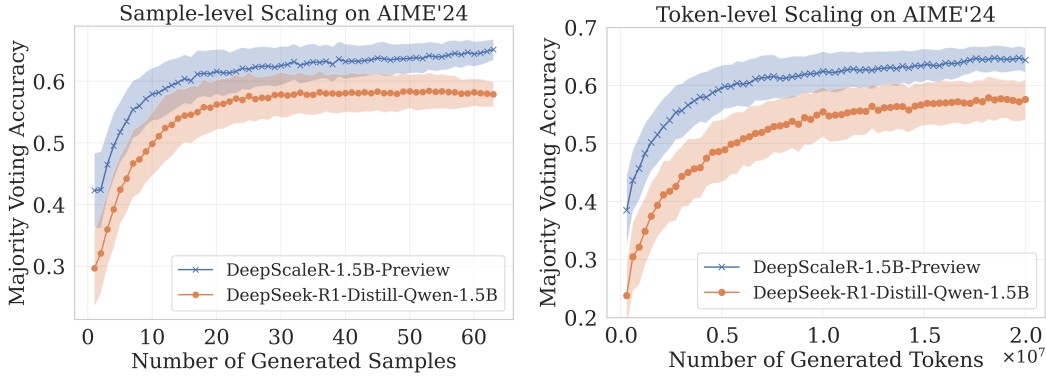

Figure 5: Test-time scaling comparison between DeepScaleR and `Deepseek-R1-1.5B-Distill`. **Left:** Mean majority voting accuracy (with standard deviation) as a function of the number of sampled responses. **Right:** Mean majority voting accuracy (with standard deviation) as a function of the total number of generated tokens. DeepScaleR consistently outperforms the baseline while requiring much fewer tokens to reach comparable accuracy.

| Max Tokens | Coverage (%) | Truncated (%) |
|---|---|---|
| 0 | 0.0 | 100.0 |
| 512 | 10.9 | 91.0 |
| 1024 | 45.5 | 53.1 |
| 2048 | 61.9 | 34.2 |
| 3072 | 66.8 | 28.8 |
| 4096 | 69.3 | 25.4 |
| 5632 | 71.1 | 19.8 |
| 7680 | 71.8 | 2.9 |
| 8192 | 71.8 | 0.0 |

Figure 6: Coverage and truncation for Qwen3-0.6B on CountDown under different cutoffs.

Figure 7: Coverage and truncation for Qwen3-0.6B on CountDown as a function of maximum context length.

lengthening technique leads to significantly more concise reasoning, which greatly improves the efficiency of test-time scaling. As shown in the right panel of Figure 5, DeepScaleR reaches 50% majority voting accuracy using 4.2M fewer tokens (equivalent to a savings of 14K tokens per problem), demonstrating a substantial reduction in inference cost.

## 4.2 CONTEXT LENGTH SCHEDULING ON THE COUNTDOWN TASK

To investigate the effect of different context length schedules and validate our principles for selecting the context window, we conduct a set of additional RL scaling experiments on the COUNTDOWN task using QWEN3-0.6B as the base model.

**Coverage analysis.** Before running experiments, we first examine the behavior of the base model under different context cutoffs. Figure 7 and Table 6 report coverage (fraction of problems solved within the cutoff) and truncation (fraction of solutions clipped). We observe a concave growth curve: coverage rises rapidly for the first 1K tokens (0% to 45.5%), then slows beyond 2K. Meanwhile, accuracy drops substantially at higher cutoffs (e.g., only 25 out of 225 problems are solved between 4K–8K), suggesting these longer problems remain unsolved.

This implies that directly scaling at 8K is inefficient, as substantial compute is wasted on truncated or incorrect long solutions that do not contribute to the gradient under GRPO. Our scheduling principles therefore suggest initializing within the 1K–2K range, where coverage gains are steepest but diminishing returns have not yet set in.

| Schedule | Start Acc. (%) | Final Acc. (%) | Avg. Len | Steps |
|---|---|---|---|---|
| $512 \rightarrow 4K \rightarrow 8K$ | 10.9 | $67.3 \rightarrow 79.6 \rightarrow 82.5$ | 1760 | $350 \rightarrow 50 \rightarrow 50$ |
| $1K \rightarrow 4K \rightarrow 8K$ | 45.5 | $70.2 \rightarrow 82.7 \rightarrow 85.4$ | 2061 | $225 \rightarrow 50 \rightarrow 25$ |
| $2K \rightarrow 4K \rightarrow 8K$ | 61.9 | $74.2 \rightarrow 80.7 \rightarrow 84.9$ | 3045 | $150 \rightarrow 50 \rightarrow 25$ |
| $4K \rightarrow 8K$ | 69.3 | $75.1 \rightarrow 81.9$ | 3054 | $75 \rightarrow 50$ |
| 8K (direct) | 71.8 | 81.1 | 3022 | 100 |

Table 3: Countdown task results under different ICL schedules. Accuracies are reported at each stage of training.

**Effects of Different Iterative schedules**   We evaluate five iterative context lengthening (ICL) schedules—(1) $512 \rightarrow 4K \rightarrow 8K$, (2) $1K \rightarrow 4K \rightarrow 8K$, (3) $2K \rightarrow 4K \rightarrow 8K$, (4) $4K \rightarrow 8K$, and (5) direct 8K—using the same dataset and RL configuration, with context switches triggered from the best checkpoint before reward plateaus. Results are presented in Table 3. Across all setups, iterative scheduling consistently outperforms direct scaling. Runs 2 and 3, which start in the 1K–2K range, achieve the highest final accuracy (85.4% and 84.9%, respectively), validating our scheduling principles; Run 2 ($1K \rightarrow 4K \rightarrow 8K$) further provides the best trade-off, reaching top accuracy with shorter outputs. In contrast, Run 1 (512 start) begins too low, requiring significantly more steps to recover; Run 4 (4K start) skips the steep-gain region, leading to weaker outcomes (81.9%); and direct 8K training (Run 5) is the least efficient, plateauing at 81.1% despite longer responses.

These results empirically validate our two heuristics—selecting the initial context near the steepest coverage gain and expanding when learning plateaus—showing that iterative scheduling serves as an implicit curriculum that yields higher accuracy, more efficient training, and more concise solutions.

## 5 KEY TAKEAWAYS

**RL scaling can manifest in small models as well**   Deepseek-R1 [12] demonstrates that applying RL directly on small models is not as effective as distillation. Their ablations show that RL on Qwen-32B achieves 47% on AIME, whereas distillation alone reaches 72.6%. A common myth is that RL scaling only benefits large models. However, with high-quality SFT data distilled from larger models, smaller models can also learn to reason more effectively with RL. Our results confirm this: RL scaling improved a distilled model's AIME accuracy from 28.9% to 43.1%! These findings suggest that neither SFT nor RL alone is sufficient. Instead, by combining high-quality SFT distillation with RL scaling, we can truly unlock the reasoning potential of LLMs.

**Iterative lengthening enables more effective length scaling**   Prior works [53, 17] indicate that training RL directly on 16K context yields no significant improvement over 8K, likely due to insufficient compute for the model to fully exploit the extended context. And a recent work [26] suggests longer response lengths consists of redundant self-reflection that leads to incorrect results. Our experiments are consistent with these findings. By first optimizing reasoning at shorter contexts (8K), we enable faster and more effective training in subsequent 16K and 24K runs. This iterative approach grounds the model in effective thinking patterns before scaling to longer contexts, making RL-based length scaling more efficient.

## 6 CONCLUSION

In this work, we introduce a novel iterative context lengthening technique for effective RL scaling. Our approach gradually expands the model's context windows during training (8K→16K→24K), stabilizing learning and encouraging concise reasoning. Leveraging this technique, we train Deep-ScaleR, a 1.5B model that achieves 43.3% Pass@1 on AIME2024— improving by 14.3% over its base model and matching OpenAI's `o1-preview` on various math reasoning benchmarks. Our ablation study shows that iterative context lengthening is more effective than direct RL scaling, and enables stronger and more efficient test-time scaling.

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

## A  APPENDIX

### A.1  TRAINING HYPERPARAMETERS  ADDITIONAL TRAINING RESULTS

| Hyperparameters | 8k | 16k | 24k |
|---|---|---|---|
| Train Batch Size | 128 | 128 | 128 |
| GRPO Group Size | 8 | 16 | 16 |
| Max Response Length | 8192 | 16384 | 24576 |
| Learning Rate | $1 \times 10^{-6}$ | $1 \times 10^{-6}$ | $1 \times 10^{-6}$ |
| PPO Mini-Batch Size | 64 | 64 | 64 |
| PPO Epochs | 1 | 1 | 1 |
| KL Loss Coefficient | 0.001 | 0.001 | 0.001 |
| Rollout Temperature | 0.6 | 0.6 | 0.6 |
| Total Steps | 1040 | 480 | 250 |

Table 4: Training hyperparameters for DeepScaleR's 8k, 16k and 24k runs.

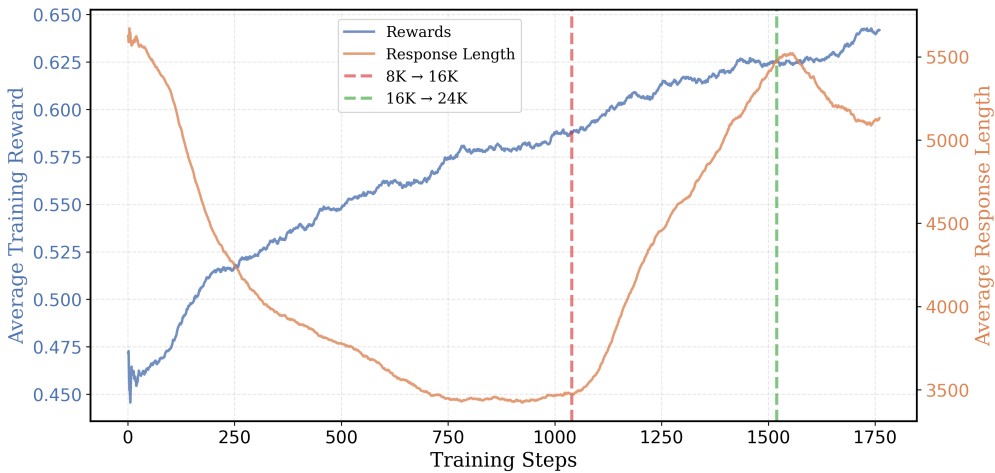

Figure 8: DeepScaleR's average response length and training rewards as training progresses. The curves shows the running average over a window size of 100.

