# OpenReview forum: "DeepScaleR: Effective RL Scaling of Reasoning Models via Iterative Context Lengthening"
_ICLR.cc/2026/Conference — Submitted to ICLR 2026_

### Official Review · Reviewer_P2gr · 2025-10-31

**Soundness:** 3
**Presentation:** 3
**Contribution:** 3
**Rating:** 4
**Confidence:** 5

**Summary:**

This approach yields substantial performance gains with dramatically reduced computational cost. DeepScaleR-1.5B achieves 43.3% Pass@1 on the AIME2024 math benchmark—a 14.3 percentage point improvement over its base model.

**Strengths:**

1.DeepScaleR-1.5B achieves 43.3% Pass@1 on the AIME2024 math benchmark.
2.The GRPO algorithm is advanced.

**Weaknesses:**

1.The performance for 1.5B-LLM is not in the sota range, for example nvidia-1,5b
2.These is not very much novelty in the proposed algorithm.
3.These is not evluation on the code dataset in the experiments as code is also a good task for the reasoning ability of LLMs.
4.There is big gap netween the performacne of the proposed LLM and the performance of the sota LLMs on 1.5B paramater scale.

**Questions:**

1.Why does the 1.5B-parameter LLM fail to achieve state-of-the-art performance, particularly when compared to models like NVIDIA's 1.5B?
2.What novel contributions does the proposed algorithm offer beyond existing methods?
3.Why is there no evaluation of the proposed method on code-related datasets, despite code being a strong indicator of reasoning ability in LLMs?
4.Why is there a significant performance gap between the proposed 1.5B-parameter LLM and current state-of-the-art LLMs at the same scale?

---

> ### Author Response · Authors · 2025-12-03
>
> We thank the reviewer for the assessment. We would like to address the concerns regarding SOTA comparisons and novelty, as there appears to be a misunderstanding regarding the timeline and the relationship between our work and the models cited.
>
> ### Comparison to State-of-the-Art & Nvidia-1.5B (Weaknesses 1 & 4)
> The reviewer notes that our model is not SOTA compared to "nvidia-1.5b" (Nemotron-Research-Reasoning-Qwen-1.5B). It is crucial to clarify two points:
> 1. **Timeline**: The Nvidia model was released after our paper work. At the time of our work was released, DeepScaleR was state-of-the-art among open-weights models in this size class.
> 2. **Impact & Attribution**: Far from outperforming our method, the Nvidia-1.5B model actually builds directly upon our work. As stated in their official release (see: [Huggingface Model Card](https://huggingface.co/nvidia/Nemotron-Research-Reasoning-Qwen-1.5B)), they utilized our open-sourced dataset (DeepScaleR-Preview-Dataset) for training and cite DeepScaleR as their primary baseline. The fact that subsequent SOTA models rely on our data curation and training recipe serves as strong evidence of our contribution's significance, rather than a weakness in our performance.
>
> ### Novelty of the Algorithm (Weakness 2)
> While we utilize GRPO as the underlying optimizer, the core novelty of our work is the Iterative Context Lengthening recipe. We show with our main experiments and ablation studies that this technique enables more efficient and effective RL scaling, resulting in a model with more concise reasoning, reduced training cost, but strong performance.
>
> ### Evaluation on Code Datasets (Weakness 3)
> While our main focus in this work is on mathematical reasoning tasks, training on math indeed shows generalization to coding domain. Our evaluation on LiveCodeBench v5 shows that the performance improves from 16.1% to 19%, despite not training on any code reasoning data. We will incorporate this results into our paper.

---

### Official Review · Reviewer_ZJL6 · 2025-11-01

**Soundness:** 3
**Presentation:** 3
**Contribution:** 2
**Rating:** 4
**Confidence:** 4

**Summary:**

The paper presents DeepScaleR, an efficient and effective training recipe for reasoning models. Specifically, it proposes high-quality data curation and iterative context lengthening, which gradually extends the context window during RL training (from 8K to 16K to 24K) to help the model learn efficient short reasoning before longer reasoning. Evaluation shows the effectiveness of context scheduling, as the trained 1.5B model achieves 43.3% Pass@1 on AIME2024, a 14.3% improvement over the base model and comparable to OpenAI’s o1-preview. This paper aims at reporting a RL-based training recipe that enables models to achieve good reasoning performance efficiently.

**Strengths:**

The paper is easy to follow.

The idea of iterative context lengthening scheduler is straightforward.

The method leads to a 1.5B model that has shows good performance in reasoning benchmarks.

**Weaknesses:**

Limitation of the model and experimental results. Although the claim is that the technique can enable small model with efficient training to have reasoning ability, the current technique is applied only to train a 1.5B model. It is currently not clear whether the length scaling can be universally effective for other model configurations, i.e. different sizes or different architectures. Furthermore, is this technique applicable for models with larger size, i.e. 7B model.

Question regarding length cutting: during training, by cutting at i.e. ctx length = 8k, do you explicitly let the model generate the final answer, i.e. by appending the final <think> token after it reaches 8k output length? How is this step done?

Question regarding the 24k ctx length ablation. Why does the plot show almost no improvement (figure 4) with static context length during training?

**Questions:**

See weakness.

---

> ### Author Response · Authors · 2025-12-03
>
> ### Generalization to larger models (Weakness 1)
> While our ablation studies focused on the 1.5B scale to allow for extensive experimental iteration, the core principle of iterative context lengthening is architecture-agnostic. We would like to highlight that recent follow-up works (e.g., DeepCoder [1], ProRL [2]) have already applied techniques similar to our length-scaling proposal and demonstrated their effectiveness at larger model sizes (including 7B+). This suggests that the curriculum learning benefit—learning efficient reasoning before long-context reasoning—transfers well to larger architectures. We will update the final manuscript to discuss these broader implications.
>
>
> ### Implementation of Length Cutting (Question 1)
> We utilize a "hard" cutoff mechanism. During the 8K training phase, we set max_tokens=8096. We do not force the model to generate a final answer if it hits this limit. Instead, if the model fails to output the solution within the 8K budget, the trajectory is truncated and because no solution is outputted, it is marked as incorrect (zero reward).
>
>
> ### Static Context Ablation / Figure 4 (Question 2)
> Regarding the static 24K context baseline in Figure 4: The plot appears flat primarily due to the scale of improvement compared to our proposed method, but the model is learning, albeit inefficiently. Specifically, the static baseline yields a ~5% improvement in test accuracy (29% $\rightarrow$ 34%) and a ~2% improvement in training accuracy (55% $\rightarrow$ 57%) over ~400 steps. The visual disparity highlights the central contribution of our paper: starting immediately with a massive search space (24K) makes optimization difficult, whereas our iterative scheduler significantly accelerates convergence and final performance.
>
> [1] DeepCoder: A Fully Open-Source 14B Coder at O3-mini Level
>
> [2] ProRL: Prolonged Reinforcement Learning Expands Reasoning Boundaries in Large Language Models

---

### Official Review · Reviewer_ky9p · 2025-11-01

**Soundness:** 2
**Presentation:** 3
**Contribution:** 2
**Rating:** 4
**Confidence:** 4

**Summary:**

The paper addresses the challenge that training large reasoning models with Reinforcement Learning is computationally expensive and believed to be ineffective for smaller models. The authors propose iterative context lengthening, a training strategy that acts as an implicit curriculum. Instead of training at a large, fixed context (e.g., 24K), iterative context lengthening starts with a short context (8K) to force the
model to learn efficient reasoning, then progressively increases the context length (to 16K, then 24K) as performance plateaus. Using this method, their 1.5B parameter DeepScaleR model achieves 43.3% Pass@1 on AIME2024, a 14.3% gain over its base model, matching o1-preview. This was achieved with a 2.6x reduction in compute cost compared to a direct 24K training baseline.

**Strengths:**

- Simple and Effective Method: Iterative context lengthening is an intuitive, simple, and highly effective training strategy that provides a more stable and efficient curriculum than direct
long-context training.
- Strong Performance and Efficiency: The 1.5B model achieves a significant +14.3% absolute
gain on AIME2024, demonstrating that small models can be scaled with RL. This is achieved
with 2.6x less training compute and results in a model that is also more efficient at inference time.
- Good Ablations: The ablation study (Figure 4) clearly proves iterative context lengthening's
superiority over a direct 24K baseline.

**Weaknesses:**

- Questionable Base Model Choice: The base model selection of a Qwen-2.5B-Math model is a concern, as this model series is known for potential test-set contamination on math benchmarks. This makes it difficult to definitively attribute the +14.3% AIME gain solely to the iterative context lengthening technique rather than the base model's pre-existing (and potentially "tainted") capabilities. The claims would be far more convincing if the authors either:
  - Replicated the experiment with a different base model (e.g., the Qwen3-0.6B used for the
COUNTDOWN task or Gemma-3-1B).
  - Evaluated the base and final trained models on a contamination-resistant benchmark,
such as LiveMathBench, to confirm the gains are from reasoning and not memorization.

- Unclear Mechanism for "Shorter Reasoning": The paper's claim that a constrained window "Encourages shorter reasoning" is not well substantiated. A constrained window merely truncates long responses, filtering them from the gradient update. This doesn't necessarily teach the model to be concise. It's plausible that the RL algorithm (GRPO, which is known to increase response length) still favors longer reasoning paths, which are then simply cut off. This would result in fewer valid, complete responses within the constrained window, not more efficient ones.
  - To truly support this claim, the authors should show that the 8K-trained model produces a
higher percentage of valid, complete responses (i.e., those reaching a final answer) within
a fixed 8K/16K/24K context than the base model. The data in Table 1, which only shows
average token length, is insufficient proof of this efficiency gain.

- Ambiguous Test-Time Scaling Evaluation: The test-time scaling analysis in Figure 5 is missing a critical detail: the maximum context window used for generating the 64 samples. It is unclear if this was capped at the 24K training limit or was uncapped.
  - A more insightful comparison, given the paper's theme, would be to evaluate how both the base model and DeepScaleR perform at test-time when the context window is expanded beyond the training limit with scaling at test time (e.g., progressively to 32K or
64K).

I am happy to increase my score if all the concerns are resolved.

**Questions:**

Please refer to the weakness.

---

> ### Author Response · Authors · 2025-12-03
>
> We thank the reviewer for the detailed feedback. We believe the following data points can directly resolve your concerns:
> ### Base Model & Contamination Concerns (Weakness 1)
> We appreciate the reviewer’s concern regarding potential contamination in the Qwen-2.5-Math lineage (our base model is DeepSeek-R1-Distill-1.5B). We note that reported contamination primarily affects the MATH500 dataset, not AIME 2024, which is our main evaluation benchmark. In fact, the original Qwen-2.5-Math model exhibits poor performance on AIME 2024 (<10% Pass@1), suggesting minimal exposure.
>
> To conclusively rule out memorization and demonstrate true generalization, we conducted an additional evaluation on AIME 2025:
>
> * Clean Evaluation: AIME 2025 was released after the training cut-off for all models considered, ensuring zero risk of contamination.
>
> * Results: Our model attains 37.1% on AIME 2025, representing a +9% absolute improvement over the base model (28.1%).
>
> These results confirm that the performance gains reported in the paper arise from genuine reasoning improvements enabled by our RL training recipe, rather than from memorization of leaked or contaminated data.
> ### Mechanism for "Shorter Reasoning" (Weakness 2)
> The reviewer asked whether the 8K constraint genuinely teaches the model to be concise, rather than merely filtering out long trajectories. To substantiate this, we measured the Truncation Ratio—the percentage of trajectories that fail to produce a final answer because they exceed the 8K token budget.
>
> * Before 8K Training: The base model exhibited a 42% truncation ratio under an 8K cap.
> * After 8K Training: The truncation ratio fell to 6%.
>
> This substantial reduction demonstrates that the model did not simply “survive” a length-based filter. Instead, it adapted its policy to reliably complete reasoning within the allowed window (94% of trajectories), confirming that the 8K constraint effectively incentivizes the discovery of more efficient reasoning strategies.
>
> ### Test-Time Scaling Context Details (Weakness 3)
> To clarify the setup for Figure 5: We utilized a maximum context window of 32K tokens during test-time generation, which matches the maximum context window of the original base model. The model was not artificially constrained to the 24K training limit during inference.

---

### Meta-Review · Area_Chair_D4SW · 2026-01-10

**Summary:**

the paper is recommended for rejection because the experimental scope is limited to a single model size, leaving significant uncertainty regarding the generality of the findings. While the rebuttal clarified some technical doubts, the core methodological novelty was perceived as limited.

**Reviewer Concerns:**

•	Addressed: Base Model Contamination: Resolved by additional evaluation on AIME 2025; Mechanism for Concise Reasoning;  SOTA Comparison

•	Outstanding:  Generalization to Larger Scales; Methodological Novelty: Concerns persist that the contribution is primarily an intuitive scheduling recipe rather than a fundamental algorithmic advancement

**Reviewer Scores:**

Reviewer ky9p: Likely would have maintained a 4.

Reviewer ZJL6: Likely would have maintained a 4.

Reviewer P2gr: Likely would have maintained a 4.

---

### Decision · Program_Chairs · 2026-01-26

Reject